# Comparison of the Oxidative Stability and Antioxidant Activity of Extra-Virgin Olive Oil and Oils Extracted from Seeds of *Colliguaya integerrima* and *Cynara cardunculus* under Normal Conditions and After Thermal Treatment

**DOI:** 10.3390/antiox8100470

**Published:** 2019-10-09

**Authors:** Diana Abril, Yaneris Mirabal-Gallardo, Aymeé González, Adolfo Marican, Esteban F. Durán-Lara, Leonardo Silva Santos, Oscar Valdés

**Affiliations:** 1Departamento de Biología y Química, Facultad de Ciencias Básicas, Universidad Católica del Maule, Talca 3605, Chile; dabril@ucm.cl (D.A.); aymee.gonzalez@synergiabio.com (A.G.); 2Vicerrectoría de Investigación y Postgrado, Universidad Católica del Maule, Talca 3605, Chile; ymirabal@ucm.cl; 3Chemistry Institute of Natural Resources, University of Talca, Talca P.O. Box 747, Chile; amarican@utalca.cl; 4Bio & NanoMaterials Lab, Drug Delivery and Controlled Release, Universidad de Talca, Talca 3460000, Chile; eduran@utalca.cl; 5Departamento de Microbiología, Facultad de Ciencias de la Salud, Universidad de Talca, Talca 3460000, Chile; 6Laboratory of Asymmetric Synthesis, Chemistry Institute of Natural Resources, University of Talca, Talca P.O. Box 747, Chile; lssantos@utalca.cl

**Keywords:** *Colliguaya integerrima* oil, *Cynara cardunculus* oil, thermal behavior, minor components, antioxidant activity

## Abstract

We investigated the potential of two oil extracts from seeds of *Colliguaya integerrima* (CIO) and *Cynara cardunculus* (CO) to use as nutritionally edible oils. For this purpose, oil quality was accessed by determining the fatty acid composition, peroxide value, acid value, iodine value, saponification number, phenolic contents, and oxidative stability during thermally induced oxidation of CIO and CO oils and compared to those of extra-virgin olive oil (EVOO). The chemical composition results demonstrated that both oils could be nutritional sources of essential unsaturated fatty acids. Moreover, according to the gravimetric analysis, the main decomposition step occurred in the temperature range of 200–420 °C, showing a similar thermal behavior of EVOO oil. However, CO and EVOO oils showed a higher phenolic content at degradation onset temperature (T_0_) in contrast with CIO oil. The antioxidant activity of the different studied oils showed a direct correlation with the phenol contents, up to temperatures around 180 °C, where the percentage of free radical scavenging assay for EVOO was higher than CO in contrast with the TPC values. Finally, we analyzed the minor components before and after heating CIO and CO at 180 °C by gas chromatography–mass spectrometry (GC–MS) using library search programs.

## 1. Introduction

Many developing countries, especially those in tropical regions, experience considerable food and nutrition problems. A potential strategy for improving food security is taking advantage of local resources to meet the needs of the growing population [1]. Specifically, oils and fats are the most used for cooking and frying in local diets. They are a major source of energy and provide essential lipid nutrients. The degradation of oil after frying generates excessive free radicals and lipid peroxidation, and this free radical production causes several pathological conditions including atherosclerosis, aging, nephrites, diabetes mellitus, rheumatic diseases, cardiac and cerebral ischemia, cancer, and adult respiratory distress syndrome [2]. 

Oxidative stability is an indispensable parameter in assessing the quality of oils and fats, and it is significantly influenced by their fatty acid composition and minor components such as tocopherols, phytosterols, vitamin E, and phenolic compounds [3]. It can be defined as a necessary period of time for the creation of secondary products, which can be identified under diverse conditions. This is known as induction time, and it results in a rapid increment in the lipid oxidation rate [4]. Phenolic compounds are very important natural antioxidants for the stabilization of unsaturated fatty acids and provide efficient protection against oxidative stress in the human body [5]. These substances are absent in all commercially refined oils due to their elimination during refining. In addition, thermal processing affects their chemical structure and potentially their antioxidant activity.

Olive oil is extremely healthy; accordingly, it is an excellent choice for use during cooking, even for high heat methods like frying due to the higher oxidation stability. This higher oxidation stability is mostly attributed to its low content of polyunsaturated fatty acids, which require a comparatively low content of vitamin E for effective protection. Cheikhousman et al. confirmed that polyphenols perform as vitamin E stabilizers during olive oil heating, which produces an adequate balance of oxidative protection between these two antioxidant families [6]. 

The *Euphorbiaceae* family is one of the largest and most diverse families of flowering plants, with around 300 genera and over 7000 species. The majority of *Euphorbiaceae* are tropical or subtropical. The majority of taxa in the family are short-lived, understory trees, shrubs, or herbs that grow characteristically in secondary vegetation or open arid regions, although some taller trees belong to the taxa as well [7]. The genus Colliguaya, specifically *Colliguaya integerrima* is one of the most widely distributed species of *Euphorbiaceae*. It grows wild in the phytogeographical provinces of Patagonia and Monte in Argentina and Chile [8]. 

In contrast, wild *Cynara cardunculus* L. var. *sylvestris* (Lamk) Fiori is a non-domesticated robust perennial plant known for its rosette of large spiny leaves, branched flowering stems, and blue-violet flowers. It belongs to the family of *Asteraceae*, tribe *Cynareae*, and is native to the Mediterranean basin, where it grows in dry and undisturbed areas [9]. The genus *Cynara*, a member of the *Asteraceae* family (also known as Compositae), has eight species and four subspecies. The members of the species *Cynara cardunculus* L. (2*n* = 2*x* = 34) are globe artichoke (var. *scolymus* (L.) Fiori), cultivated cardoon (var. *altilis* DC.) and the wild cardoon (var. *sylvestris* (Lamk) Fiori). The three *Cynara cardunculus* varietal botanicals are fully cross-compatible, and their F1 hybrids are fully fertile [10]. Some genus of *Cynara cardunculus* are cultivated for industrial applications, specifically energy purposes [11], distinct from the *Colliguaya integerrima* which grows wildly. 

In addition, seeds of *Cynara cardunculus* and *Colliguaya integerrima* have significant potential as new sources of oil because of the biological compounds contained, such as phenylpropanoids (mono- and di-caffeoylquinic acids), sesquiterpene lactones, among others, which may make these oils have special advantages [11,12,13].

The purpose of the present study was to evaluate the chemical composition and oxidative stability during thermally induced oxidation of two oils extracted from seeds of *Colliguaya integerrima* (CIO) and *Cynara cardunculus* (CO); the study compared these two oils with extra-virgin olive oil (EVOO) as a nutritional edible oil among animal fats and vegetable oils. To our knowledge, no studies have been previously reported about the oil content of *Cynara cardunculus* and *Colliguaya integerrima* or their possible use as cooking oil.

## 2. Materials and Methods

### 2.1. Chemicals and Reagents

Methanol, cyclohexane, sodium thiosulfate pentahydrate, potassium hydroxide (KOH), methylene chloride (CH_2_Cl_2_), and *n*-hexane (HPLC purity grade) were supplied by Arquimed (Santiago, Chile). 1,1-Diphenyl-2-picrylhydrazyl (DPPH), iodine monochloride solution (Wijs reagent), Starch solution, Phenolphthalein, and Folin-Ciocalteu reagent were purchased from Sigma-Aldrich (St. Louis, MO, USA). The extra-virgin olive oil (Olivares de Quepu) in dark green glass bottles was obtained from the local market; it is produced in the Maule region of Chile. Other reagents used were of analytical grade. Finally, Supelco^®^ 37 component fatty acid methyl ester (FAME) mix in dichloromethane (varied concentrations) for GC-MS analysis was purchased from Sigma-Aldrich (St. Louis, MO, USA).

### 2.2. Colliguaya integerrima and Cynara cardunculus Seeds

The seeds of *Colliguaya integerrima* and *Cynara cardunculus* were collected in February 2016 at Pehuenche International Pathway (Los Cóndores) at the foothills of Talca (19 H 348271 6026739) and Curepto, (18H 767945, 80 E; 6118448,36 S), respectively. All seeds were dried at room temperature, grounded in a BIOBASE MPD-102 Disintegrator (Shanghai, China), and their particle size was classified by sieving (20 mesh size). The seeds used in the extraction of the oils from *Colliguaya integerrima* and *Cynara cardunculus* consisted of 35% and 38% of the mass retained in the mesh sieve, respectively.

### 2.3. Vegetable Oil Extraction

Oil extraction was performed according to the AOAC method Am2-93 [14]. Using a Soxhlet apparatus and *n*-hexane (150 mL) as an extraction solvent, about 250 g of *Colliguaya integerrima* seeds were extracted for 8 h. Then, by distillation under reduced pressure at 40 °C, *n*-hexane was removed. The orange oil obtained from *Colliguaya integerrima* was stored at 4 °C under nitrogen atmosphere until further study. The extraction for each harvested sample was performed in triplicate, obtaining a 27.3% ± 2.9 yield. The same procedure was used for the *Cynara cardunculus*; however, the oil obtained had a yellow tint with 23.0% ± 2.5 of yield. Figure 1 shows a photograph of the plants, the obtained oil, and seeds of *Colliguaya integerrima* (a) and *Cynara cardunculus* (b).

### 2.4. Oil Characterization

#### 2.4.1. Specific Extinction Coefficient (*K_270_* and *K_232_*) of Extra-Virgin Olive Oil 

Extinction coefficient (*K*_270_, *K*_232_ and Δ*K*) determination were carried out following the analytical methods described by Harhar et al. with slight modifications [15]. Specifically, the oil samples obtained from extra-virgin olive (EVOO) was diluted in cyclohexane until obtaining 1% (*w/v*) solution, respectively. Then, oil samples were measured and matched using synthetic fused silica cuvettes running a solvent blank as a reference. Absorption measurements for purity determination were taken at 232, 266, 270, and 274 nm in a UV spectrophotometer (Thermo Spectronic Genesys 10 UV). K values were calculated according to Equations (1) and (2).
(1)$$K_{\lambda }=\frac{\mathrm{Abs}\;\lambda }{\mathrm{DxL}}$$
(2)$$\Delta K=K_{270}-\frac{K_{266}+K_{274}}{2}$$
where Abs λ is the absorption, D is the dilution expressed in g/L; L is the cuvette pathlength and K_λ_ is the specific extinction coefficient at different wavelengths. 

#### 2.4.2. Acid Value

The acid value (AV) of the obtained oils was determined according to the Association of Official Analytical Chemists (AOCS) Official Method Cd 3d-63 [16]. First, a mixture solution of ethanol–ethyl ether (1:1 ratio) neutralized with 0.1N KOH solution was prepared. Then, 5 g of EVOO, CIO, and CO oil samples were solubilized using this neutral solution; 2 mL of phenolphthalein indicator were also added. Titrations of each oil sample were performed using 0.5 N alcoholic KOH solution until the consistent appearance of pink color. The results were expressed in 1 mg of KOH/g of oil.

#### 2.4.3. Iodine Value

The iodine value (IV) was measured according to the AOAC Official Method 920.158 (Hanus method) [17]. To determine IV, 0.2 g of EVOO, CIO, and CO oil samples were dissolved in cyclohexane, then 25 mL of Wijs reagent 0.2 N was added in the dark. After 30 min, 10 mL of 15% potassium iodide (KI) solution was added. The mixture was titrated with a 1 N solution of sodium thiosulfate pentahydrate under constant and vigorous shaking. We used a fresh 1% starch solution as an indicator. The results were expressed in g of I_2_ per 100 g of oil.

#### 2.4.4. Peroxide Value

The peroxide value (PV) was determined according to AOAC Official Method 965.33 [17]. To determine PV, 5 g of EVOO, CIO, and CO oil samples were dissolved in 30 mL of acetic acid/chloroform solution (3:2 *v/v*). After adding saturated KI solution (0.5 mL) to oil solution, the mixture was titrated with a standard solution of sodium thiosulfate (0.01 N) under magnetic agitation. The blank also was determined under similar conditions using the fresh starch solution as an indicator. The results were expressed as milliequivalents O_2_ per kg of oxidation oil.

#### 2.4.5. Saponification Number

The saponification number (SN) was measured according to the AOAC Official Method 920.160 [17]. To measure the SN, 3 g of EVOO, CIO, and CO oil samples were dissolved in 0.7 N alcoholic KOH solution and the mixture sample was refluxed for complete saponification. After 2 h, the solution was titrated with a 1 N sulfuric acid solution using a 1% phenolphthalein ethanolic solution as an indicator. A blank determination was conducted simultaneously with the sample. 

#### 2.4.6. Fatty Acid Composition

The fatty acid profile was determined as fatty acid methyl esters by gas chromatography–mass spectrometry (GC–MS). The methyl esters were prepared using the method described by Morrison and Smith [18]. The separation of the fatty acid esters was performed using QP 5000 Shimadzu (Kioto, Japan) gas chromatographer with a mass spectrometer and autosampler was used as well as the 1.2 Class-5000. A fused-silica column coated with the DB-5 stationary phase was utilized (30 m × 0.2 mm inner diameter, a dry film thickness of 0.25 µm, J & W Scientific). The initial oven temperature was 60 °C, which was kept for 5 min; a 2 °C min^−1^ temperature increase was programed until it reached 220 °C; this temperature was kept for 30 min. The injector temperature was 220 °C. Helium was used as a carrier gas with a 1.0 mL min^−1^ flow. The injection volume was 1 µL (1% solution in CH_2_Cl_2_) with a 1:10 split ratio. Column pressure was 100 kPa. Mass detector conditions were the following: source temperature, 240 °C; electron impact mode (EI), 70 eV; scan rate of 1 scan s^−1^, and acquisition range, 29–450 u. 

Components were identified by comparing retention times related to a linear standard made with Supelco^®^ 37 (Sigma-Aldrich, Santiago, Chile) Component FAME Mix in dichloromethane (varied concentrations) of an alkane series (C9-C24) and their mass spectra to those from the Wiley 330000 database and reviewed from the literature.

#### 2.4.7. Calculated Oxidizability Value

The calculated oxidizability (Cox) value of the oils was calculated by applying the formula (3) proposed by Fatemi et al. [19]:(3)$$\mathrm{Cox}=\frac{\left[1\left(16:1\%+18:1\%+20.1\%\right)+10.3\left(18:2\%+20:2\%\right)+21.6\left(18:3\%\right)\right]}{100}$$

#### 2.4.8. Total Phenolic Content

The total phenolic content (TPC) of the hydro-alcoholic (1:1, *v/v*) fraction of the obtained oils was measured according to the Folin–Ciocalteu method [20,21]. First, 5 mL of EVOO, CIO, and CO oil samples were added to 5 mL of methanol into the ultrasound device at room temperature for 30 min to obtain the methanolic extracts. Then, approximately, 20 μL of the different methanolic extract oils were mixed with 1.58 mL of water and 100 μL of freshly diluted Folin-Ciocalteu reagent in water (1:9 *v/v*) in the dark. Then, the reaction mixture was preincubated for 8 min; then, 300 μL of sodium carbonate 20% was added and the mixture was allowed to stand for 5 min at room temperature prior to vortexing. Finally, the absorbance was obtained in a spectrophotometer (Thermo Spectronic Genesys 10 UV), at a wavelength of 765 nm after each tube was incubated for 2 h at room temperature. The assay was done in triplicate, and the TPC results were expressed as gallic acid equivalents (GAE) in milligrams per kilograms of oil. Methanolic solutions containing gallic acid at concentrations of 0, 100, 200, 300, 400, and 500 μg/mL were used for the construction of a standard curve to calculate TPC content of the oil samples according to the regression Equation (4):(4)$$A=8.2363\times 10^{-4}c+1.4907\times 10^{-2}$$
where *A* is absorbance and *c* is the concentration of GAE.

#### 2.4.9. Using Free Radical Scavenging Assay to Determine Antioxidant Activity

Using DPPH as the free radical model according to the method adapted from Brand-Williams et al. [22] and Molyneux, the scavenging activity of the hydro-alcoholic (1:1, *v/v*) fraction of the oils was estimated [23]. First, a methanolic solution of DPPH radicals was prepared at a concentration of 20 mg/L in the dark. Then, 2940 µL of methanolic solution of PPH radicals were mixed with 60 µL of methanolic oil extract and shaken vigorously and left at room temperature for 30 min. The procedure to obtain the CIO, EVOO, and CO methanolic extracts were as described in Section 2.4.3. The control was prepared using the indications above without any extract. For the baseline correction, MeOH was used and the measurements was performed at 515 nm. The free radical scavenging activity was calculated as the following Equation (5):(5)$$\%\mathrm{scavenging}\;\mathrm{DPPH}\;\mathrm{free}\;\mathrm{radical}=100\;\times \;\left(1\;-\;\frac{\mathrm{AE}}{\mathrm{AD}}\right)$$
where AE is the absorbance of the solution after adding the extract or fraction and AD is the absorbance of the blank DPPH solution. Quercetin was used as a reference compound.

### 2.5. Thermal Analysis

Oil analysis was conducted in a thermogravimetric analyzer TGA-Q500 by TA-instruments with 10 °C min^−1^ constant heating rate. The heating was 30–600 °C in synthetic air, with a flow rate of 60 mL min^−1^. Approximately, 10 mg of the obtained oils were used for all thermogravimetric analyses. The most suitable temperatures were selected after the TG/DTG curve analyses to monitor the process of oil degradation, as proposed by Forero-Doria et al. [4]. Specifically, the chosen temperatures were room temperature 30 °C (T_30 °C_), an intermediate temperature between room temperature and the start of degradation 150 °C (T_int_), the initial degradation temperature 250 °C (T_0_), the temperature corresponding to 5% degradation 273 °C (T_5%_), and the temperature corresponding to 10% degradation 275 °C (T_10%_).

### 2.6. Heating of Oils

Two different heating processes were performed for the studied oils. The first heating process was performed to determine the total phenolic content and antioxidant activity of the oils for every selected temperature obtained in the thermal analysis. Specifically, 1 mL of each samples was added to a clean, acid-washed glass beaker (outer diameter about 18 mm, capacity 5 mL) and heated in a hot place with a thermometer to control the temperature. Each sample was subjected to the different heating temperatures (T_30 °C_, T_int_, T_0_, T_5%,_ and T_10%_) for 15 min. The second heating process was performed to measure the minor components present in the studied oils. In this case, we used the same methodology discussed above, but the samples were heated at 180 °C for 30 min. This temperature value simulates the conditions of the frying process. After that, the samples were directly infused into the spectrometer before and after heated at 180 °C to determine the minor components using the GC–MS technique.

### 2.7. Determination of Other Minor Components Before and After Heating at 180 °C

The GC–MS analysis of minor components before and after heated at 180 °C was obtained on a Trace 1300 spectrometer (Thermo Fisher Scientific, Waltham, MA, USA) coupled to a simple quadrupole ISQ mass Spectrometer (Thermo Fisher Scientific, Waltham, MA, USA) with an AS 3000 autosampler. The column was a Restek Rtx-5MS w/integra-guard (30 m, 0.25 mm ID, 0.25 μm ft). The working conditions were as follows: the carrier gas was helium with a flow rate of 1.5 mL min^−1^ and the temperature programmed as above. Screening of minor components was performed using the automatic RTL screener software in combination with the NIST’11 library.

### 2.8. Statistical Analysis

An ANOVA was performed to determine if there were statistically significant differences between the samples of oils under heating, as well as between the physicochemical parameters of the oils, compared with the olive oil. In both cases, Tukey HSD was used at 95% confidence level. The reported results were expressed as mean and standard deviation (SD) of triplicate samples. The software used was StatGraphics Centurion XV for Windows (StatPoint, Inc., USA) [24] and Microcal Origin 8.6 (Originlab, USA) [25].

## 3. Results and Discussion

The chemical composition and characteristics of extra-virgin olive, *Colliguaya integerrima*, and *Cynara cardunculus* oils used are shown in Table 1. The highest percentage of saturated fatty acids (SFA; mainly palmitic acid, 16:0) was found in EVOO (18.34%), CO (13.43%), and CIO (12.63%), respectively. In the case of monounsaturated fatty acids (MUFA; mainly oleic acid, 18:1), the percentages of MUFA were 28.90%, 15.07%, and 64.28% for CIO, CO, and EVOO, respectively. Furthermore, the high percentage value of polyunsaturated fatty acids (PUFA; mainly linoleic acid, 18:2) was observed in CO (71.53%), followed by CIO (58.46%) and EVOO (16.21%). If we make a comparison with the vegetable EVOO, we can observe that the CIO and CO oils were richer in polyunsaturated fatty acids.

In addition, the distributions of fatty acids in both oils (CIO and CO) are similar to others that are widely consumed, such as corn, soybean, sunflower, and canola oils. The comparison with these oils is centered in the percentage content of palmitic, oleic, and linoleic acid. The soybean oil contains 11%, 23%, and 53% [26]; sunflower oil presents 7%, 26%, and 65% [27]; corn oil contains up to 11%, 25%, and 60% [28]; and canola oils 6.6%, 15.8%, and 56% [29] of palmitic, oleic, and linoleic acid, respectively. These results demonstrate that both studied oils can be an excellent nutritional source of the essential unsaturated fatty acids. MUFA has vital functions in the structure of the cell membrane and in metabolic processes and provide nutritional benefits, conferring oxidative stability to the oils used in food processes [30]. On the other hand, long-chain polyunsaturated fatty acids (PUFA) are considered indispensable nutrients for providing adequate growth, promoting health, and preventing disease in humans, such as omega-3 PUFA [31,32].

Finally, the values of oxidizability, iodine, saponification number, and acid value are presented in Table 1. As previously mentioned in the literature, saturated fat is known as the most detrimental to human health because it increases LDL cholesterol which results in heart disease [33]. For that reason, we measured the iodine value (IV), which is related to the Cox value. The results for CIO and CO showed values of IV (124 and 125 g I_2_/100 g, respectively) and Cox (9.35 and 7.57, respectively) which were higher than those of EVOO (79.3 g I_2_/100 g for IV and 2.38 for Cox), providing it a particular resistance to oxidation.

On the other hand, the saponification value (SN) found in the oil was also studied. The SN values for CIO, CO, and EVOO were 190, 196, and 194 mg KOH/g oil, respectively, which are in the average SN range of 175–250 mg/g reported for household vegetable oils [34]. These values do not show significant differences, which means that the fatty acid composition of the studied oils is similar in molecular weight. 

The highest acid value was found for CO and CIO; however, all AV for the researched oils were less than 1.0 mg KOH/g oil, as seen in Table 1, indicating that oils did not undergo to hydrolytic processes. Additionally, the peroxide value (PV) of CIO, CO, and EVOO was less than 20 meq/kg, respectively, indicating that these oils were unoxidized and of high initial quality, as seen in Table 1.

To verify the quality of the extra-virgin olive oil, specifically the degree of deterioration, we measured the characteristics of the absorption bands between 200 and 300 nm, using UV spectroscopy. We chose this range of wavelength because the primary oxidation compounds (peroxides and hydroperoxides with two conjugated double bonds) absorb at almost 232 nm, whereas the secondary oxidation products, such as aldehydes and ketones, absorb at wavelengths of 266, 270, and 274 nm [35]. For that reason, low absorption in this region is indicative of the oil quality. We focused the discussion on values determined spectrophotometrically at 232 and 270 nm, and expressed as *K_232_*, *K_270_*, respectively, because these parameters are those that are commonly used. The results obtained for EVOO at the wavelength previously commented were 2.23 and 0.17 for *K_232_* and *K_270_*, respectively. If we compare these results with those reported by the EU Regulation 2015/1830 (≤2.50 and ≤0.22 for *K_232_* and *K_270_*, respectively), which establishes the standard method for measuring extra-virgin olive oil purity, the EVOO possesses the “extra” category showing a correct alignment with the EU Regulation limits [36].

The influence of the structure and composition on thermal degradation of the oils was verified using the information provided by thermogravimetric techniques. Thermogravimetric analyses represent a strong technique for processing raw data from multi-heating rate conditions with the goal of obtaining several physico-chemical properties related to their oxidation and stability behavior. The thermal analyses of the samples were undertaken in an oxidative atmosphere within the interval of 30–600 °C. The mass loss (TG) and derivate (DTG) curves are presented in Figure 2 and show a comparison for all studied oils. The first and most important step of thermal decomposition of commercial edible oils is the thermal stability study because the decomposition of unsaturated fatty acids starts here. According to the TG curves shown in Figure 1, the stabilities in the air are similar and the order observed was CO ≥ EVOO > CIO. A plateau was also observed, indicating thermal stability of the materials up to 200 °C with the exception of CIO oil. This parameter is very important since oils in Latin American cuisine are used mostly for stir-frying and deep-frying. The cooking methods differ in duration as well as the amount of oil used, but the high temperatures used are common in both cooking methods. Generally, temperatures typically between 160 and 191 °C produce a fried quality [37].

Analyzing the curves in Figure 2, the absence of any step of mass loss under 150 °C was observed, revealing the limited content of water that is usually available in vegetable oil [38]. Furthermore, the thermal oxidative decomposition processes occurred as several consecutive and simultaneous steps of mass loss in the range 200–600 °C. Specifically, the thermogravimetric curve of CIO exhibited six decomposition stages (98% weight loss), the first one between 200 and 300 °C, with 12% of mass loss caused presumably by the presence of Alpha-linolenyl chains (18:3). Santos and Souza [39] compared the thermal stability of different commercial edible oils using the thermogravimetric technique and concluded that thermal stability of oils depends on the composition of fatty acids and other factors. Specifically, oils with high levels of polyunsaturated fatty acids such as linoleic and linolenic acid will be susceptible to thermal deterioration. Thus, it was expected that CO would be less susceptible to thermal deterioration than CIO. The main and biggest oil decomposition occurred at 300–365 °C, with 36% of mass loss and the third at 365–390 °C, with a mass loss of 14%. The degradation mechanism probably corresponded to polyunsaturated fatty acids decomposition. The fourth and fifth decomposition steps were between 390 and 418 °C with mass loss of 23% and between 418 and 455 °C with mass loss of 10% was due to monounsaturated and saturated fatty acid decomposition, respectively. Finally, the last step had a weight loss centered in 528.52 °C, with a depreciable mass loss of fewer than 5%, corresponding to the oxidation of the carbonaceous residue.

In addition, the overall decomposition process for the other two oils (CO 96% and EVOO 98%) have in common four degradations steps in the region commented above, which is reflected as a four peak in the DTG curve. The degradation mechanisms for CO and EVOO oils are similar to those explained for the CIO. In the case of CO, the first decomposition stage between 285 and 366 °C with 41% of mass loss, the second one between 366 and 382 °C with 16% of mass loss, the third between 382 and 461 °C with 34% of mass loss and a residue of approximately 5% at 524.40 °C. Furthermore, the first decomposition stage for EVOO oil appeared between 260 and 36 °C with 41% of mass loss, the second between 360 and 398 °C with 32% of mass loss, the third between 395 and 450 °C with 21% of mass loss and a residue of approximately 4% at 525.72 °C.

The obtained data agree with the results reported by Santos et al. [40] and Dweck et al. [41]. In both studies, the thermal decomposition process for commercial oils occurs in three steps, probably corresponding to polyunsaturated (200–380 °C), monounsaturated (380–480 °C), and saturated (480–600 °C) fatty acids decomposition and the carbonaceous residue, respectively. Finally, comparing the content of unsaturated fatty acids obtained by the chemical composition shown in Table 1 (e.g., CIO, CO, and EVOO present values of 87.36%, 86.6%, and 78.72%, respectively) with the around 80% of weight loss in the first three steps discussed above, we conclude that the method is reliable.

As explained above, Latin American cuisine contains many fried dishes that are made at different temperatures. Phenolic compounds are known for having essential compounds that contribute to the antioxidant activity of oils [42]. To further understand the relation between antioxidant capacity and total phenolic content, we studied the thermal stability related to the phenolic content and antioxidant activity of CIO, CO, and EVOO oils using five temperatures chosen from the TG curves, as seen in Figure 3. Because the thermal behavior was similar among the studied oils, the temperatures selected were 30 (T_30_), 150 (T_int_), 250 (T_0_), 273 (T_5%_), and 275 °C (T_10%_), respectively.

Figure 3 shows the total phenolic content (TPC) of the different oils at each studied temperature. Analyzing Figure 4, the CO and EVOO oils had the highest TPC at 30 °C (approximately 500 mg/kg), followed by CIO oil with TPC of 289.11 mg/kg. Furthermore, the same behavior was observed when different oils were heated at T_int_ and T_0,_ showing TPC values of 428.60, 404.98, 192.80, and 400.41, 345.94, and 168.82 mg/kg for CO, EVOO, and CIO oils, respectively. However, if we continued to heat the oils, we found that the TPC was drastically affected, with CIO oil being the most affected, with values of 100.52 and 16.50 mg/kg for T_5%_ and T_10%_, respectively. The TPC value for EVOO oil (106.23 mg/kg) at T_10%_ was higher than for CO oil (70.00 mg/kg), but is almost negligible. It was thought that this behavior was because the chemical structure of the polyphenols and others minor components presents in the CO oil differed from those present in the EVOO oil, which are hydroxytyrosol, tyrosol, and their derivatives (secoiridoids) [43]. Apart from phenols and secoiridoids, triterpenes, such as maslinic and oleanolic acid, the existence of other minor components has been widely documented, contributing to the antimicrobial, anti-inflammatory, and hypoglycemic effects and contribute to the antioxidant properties found in EVOO oil [44,45]. The initial degradation temperature for the three oils studied was slightly higher than the temperature used in the frying process. For this reason, we analyzed the TPC curve for CIO, CO, and EVOO oils at the average temperature of the frying process (around 180 °C) shown in Figure 4. The results indicated that the TPC content followed the same order obtained for T_0_: CIO > EVOO > CIO.

The in vitro antioxidant activities of the different oils were analyzed using DPPH assays, as seen in Figure 5. The DPPH assay is the most common method of antioxidant evaluation and it measures the ability of a compound to transform labile H-atoms to radicals. Figure 5 shows the antioxidant capacity of the different oils at each temperature studied. It is important to highlight the direct correlation between phenol contents and antioxidant activities of oils until the temperature reached 183 °C, at which point the values of DPPH for EVOO were higher than CO oils in contrast with the TPC values. This behavior could be due to the better thermal stability of EVOO oil. Specifically, in the T 30 °C, it was observed that CO oil had the highest percentage of inhibition of the DPPH (79.8%) in contrast with EVOO (76.5%) and CIO (36.9%) oils. Also, it was observed that the antioxidant activity remained in the same order despite the decrease at T_int_ with values of 65.2%, 63.8%, and 1.5%, respectively. Finally, the antioxidant effect dropped drastically at temperatures T_0_, T_5%_, and T_10%_, respectively.

The decrease in the antioxidant activity after heating at high temperatures may be due to the components present in the oil. For example, Mba et al. studied the thermo-stability of natural tocopherols, tocotrienols, and total carotenoids in different types of oils and their blends (1:1 *w/w*) at different frying temperatures [46]. They reported that the thermo-stability of the endogenous tocochromanols and carotenoids of the studied oils and their blends during deep-fat frying affected the complex interplay of the fatty acid composition, the oxidizability of the oils, the type and quantity of phytonutrients present, and their antioxidant activity. Additionally, more polar phenolic compounds and several chemical reactions occur during the oxidation, resulting in the formation of hydroperoxides, hydrolysis, polymerization, and chemical decomposition [47]. On the other hand, Cheikhousman et al., reported that some compounds such as hydroxytyrosol and their derivatives, with a 50% reduction in extra-virgin olive oil, are extensively lost after frying fresh potatoes for only 10 min at 180 °C [6].

In consideration of the previous comments and taking into account the obtained results that the amount of total phenolics plays an important role in the antioxiative properties of CO, as demonstrated in DPPH assays, we studied the minor components in both oils (CIO and CO) in an attempt to obtain a full characterization for future applications of CO as edible oil. Thus, the identification of the minor components in the studied oils using the GC–MS technique. The GC–MS chromatograms of the CIO and CO oils before and after heated at 180 °C are shown in Figure 6. The GC–MS for EVOO oil was not reported in this study because there are already several reports of their minor components described in literature [44,45]. The time and temperature were chosen using the results reported by Mba et al., Andrikopoulos et al., and Naz et al., respectively [46,48,49].

Differences in the GC-MS chromatograms obtained for CIO and CO before and after heating up to 180 °C can be seen. When comparing CIO with CO oils, it was observed that CO achieved less oxidation after heating, which is in accordance to its components and that CIO showed more oxidable linolenic acid in its composition, as seen in Table 1. Furthermore, in an attempt to understand the composition of the minor components of CIO and CO oils, we analyzed those minor compounds by GC-MS as shown in Figure 6. GC-MS analysis demonstrated that the heating process did not significantly affect the chromatographic profile of the oils, as seen in Figure 6a–d. A total of 25 and 24 compounds were identified in Figure 6 through GC-MS for CIO and CO before and after the heating process, respectively (See supporting information, Appendix A). The major compounds found by GC-MS for the studied oils were n-Hexadecanoic acid, (Z,Z)-9,12-Octadecadienoic acid, cis-13-Eicosenoic acid, Stigmasterol, campesterol, ϒ-sitosterol, oleic acid, ϒ-tocopherols, β-amyrins, among others. As can be seen in Appendix A, the main difference between CIO and CO oils were the concentrations of the minor compounds after heating, which decreased from 51.0% to 46.5% and 37.2% to 32.5%, respectively, for CO and CIO after being heated at 180 °C. As seen in Appendix A, several of those minor compounds that were identified have other important applications in medical, cosmetic, and food fields. For example, different works reported that stigmasterol might be useful in the prevention of some kinds of cancers, including ovarian, prostate, breast, and colon. Additionally, it also possesses hypoglycemic and thyroid inhibiting properties; amyrin was used as a natural sweetener; campesterol has anti-inflammatory and antimicrobial properties and tocopherols form vitamin E which enhances the immune system and metabolism, reducing the risk of cancer and cardiovascular diseases and prevents cataracts [50,51,52].

## 4. Conclusions

The by-products extracted from seeds of *Colliguaya integerrima* and *Cynara cardunculus* could be of great economic and social importance in all countries in which it grows. Specifically, the *Colliguaya integerrima* and *Cynara cardunculus* seed oils presented unsaturated fatty acids content of 87.36% and 86.60%, respectively. In addition, the studied oils presented a similar thermal behavior to olive oil, proven in the TG–DTG curves in which the *Cynara cardunculus* seed and extra-virgin olive oil had the highest total phenol content, followed by the *Colliguaya integerrima* seed oil. Moreover, the study confirmed the relationship between the thermal stability and the phenolic content with the antioxidant activity. This correlation was maintained throughout the thermal degradation process, proving a greater content of phenolic compounds in the oil extracted from the *Cynara cardunculus* seed. However, this direct correlation was observed until the temperature reached 183 °C, at which point the antioxidant activity values for extra-virgin olive oil were higher than the oil extracted from *Cynara cardunculus* seed in contrast to the total phenol content values. Finally, a total 25 and 24 compounds were identified through GC–MS for CIO and CO before and after being heated at 180 °C, respectively. GC–MS analysis demonstrated that the heating process did not significantly affect the chromatographic profile of the oils. These preliminary results can be useful for recommending *Cynara cardunculus* seed oils as a source of safe and effective natural antioxidants, which can be used in food systems even at high processing temperatures. On the other hand, although the oil extracted from *Colliguaya integerrima* seeds did not show high antioxidant capacity, its high content of omega-3 fatty acids makes it an ideal candidate to use as a dietary supplement at low temperatures.

## Figures and Tables

**Figure 1 antioxidants-08-00470-f001:**
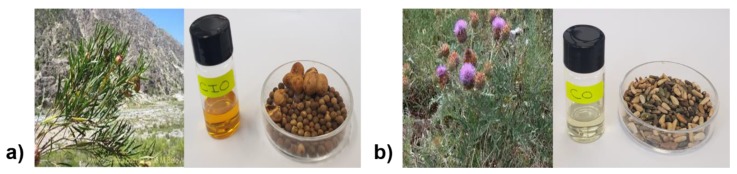
Photograph of plants, obtained oil, and seeds of *Colliguaya integerrima* (CIO) (**a**), and *Cynara cardunculus* (CO) (**b**).

**Figure 2 antioxidants-08-00470-f002:**
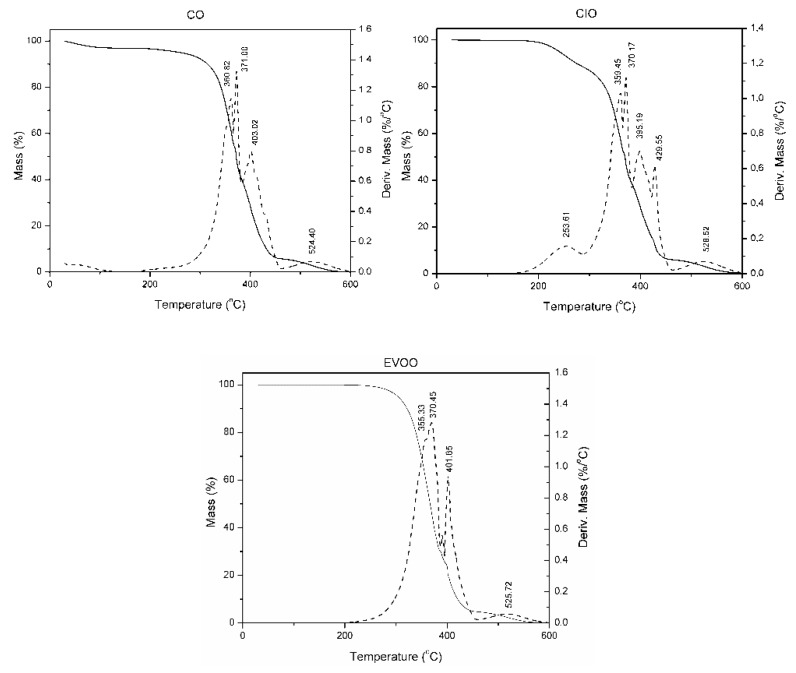
Mass Loss-Derivate (TG/DTG) thermograms of *Colliguaya integerrima* (CIO), *Cynara cardunculus* (CO), and extra-virgin olive (EVOO) oils.

**Figure 3 antioxidants-08-00470-f003:**
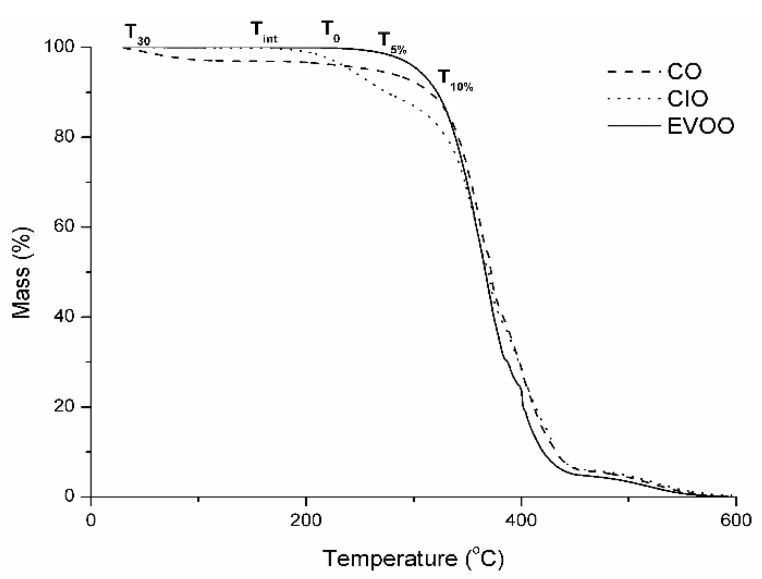
TG curve comparisons for the three different oils showing the selected temperatures corresponding to the degradation of *Colliguaya integerrima* (CIO), *Cynara cardunculus* (CO), and extra-virgin olive (EVOO) oils.

**Figure 4 antioxidants-08-00470-f004:**
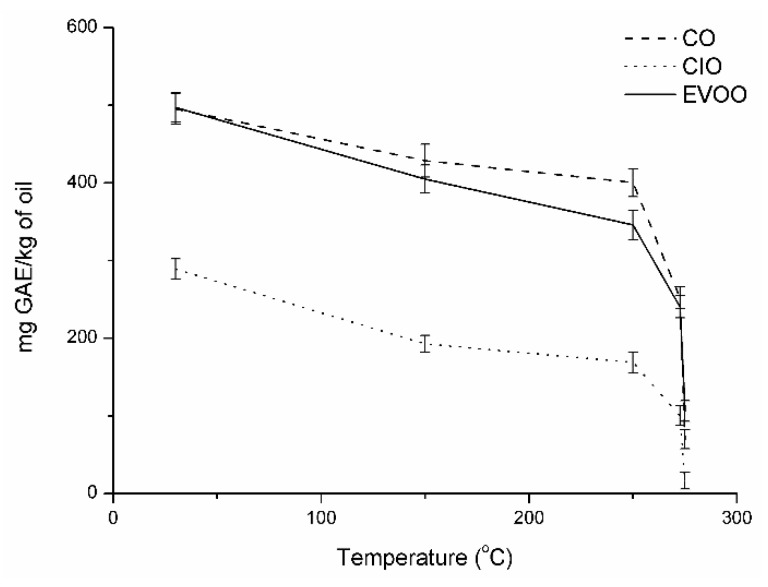
Total phenolic content of *Colliguaya integerrima* (CIO), *Cynara cardunculus* (CO) and extra-virgin olive (EVOO) oils at each studied temperature.

**Figure 5 antioxidants-08-00470-f005:**
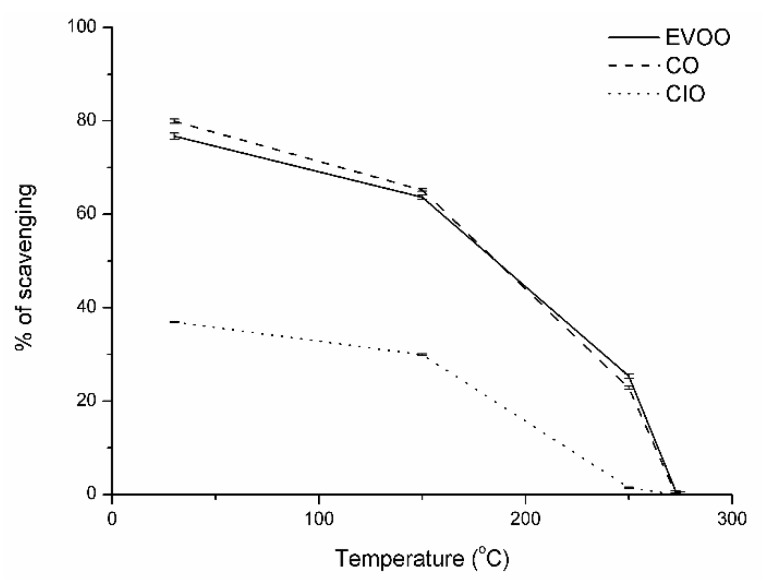
Antioxidant capacity of *Colliguayaintegerrima* (CIO), *Cynara cardunculus* (CO), and extra-virgin olive (EVOO) oils at each studied temperature.

**Figure 6 antioxidants-08-00470-f006:**
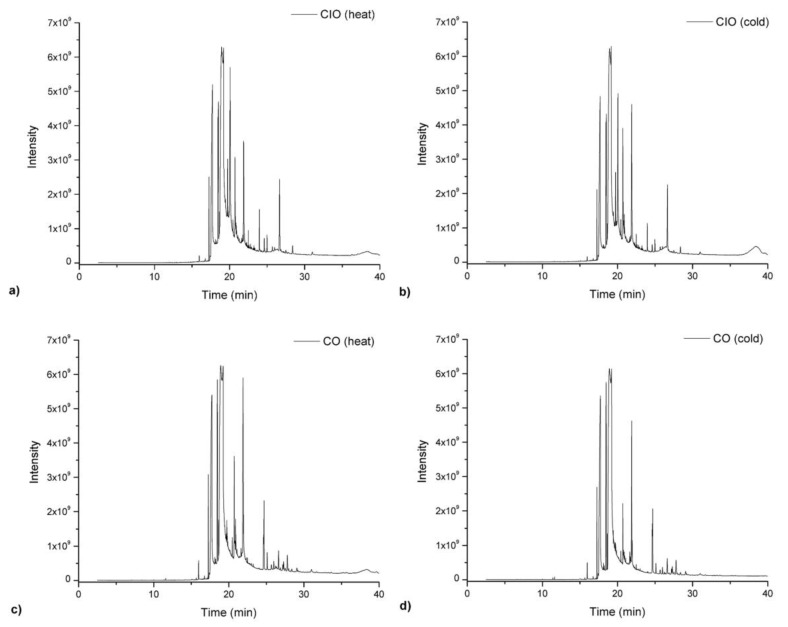
GC–MS chromatogram of CIO and CO after (**a**,**b**) and before (**c**,**d**) heating process, respectively.

**Table 1 antioxidants-08-00470-t001:** Chemical composition and fatty acid composition of *Colliguaya integerrima* (CIO), *Cynara cardunculus* (CO), and extra-virgin olive (EVOO) oils. The results were expressed as mean values ± SDs (*n* = 3). Values in the same Same letter (a,b,c) beside standard deviation (SD) in the same row indicate no statistical differences between the oils, using Tukey HSD (at 95% level of confidence).

Parameters		Oils	
	CIO	CO	EVOO
Cox value	9.35	7.57	2.38
AV [mg KOH/g oil]	0.2 ± 0.01a	0.2 ± 0.02a	0.4 ± 0.09b
IV [g I_2_/100 g oil]	124 ± 4.68b	125 ± 3.89b	79 ±1.68a
PV [meq O_2_/kg oil]	19 ± 1.29c	15 ± 0.21b	3 ± 0.84a
SN [mg KOH/g oil]	190 ± 5.21a	196 ± 3.62a	194 ± 2.01a
TPC [mg GAE/kg oil]	289 ± 13a	495 ± 19b	497 ± 19b
Fatty acid (%)			
Myristic, C_14:0_	0.06 ± 0.01a	0.12 ± 0.09a	0.01 ± 0.01a
Palmitic, C_16:0_	10.54 ± 0.35a	10.42 ± 0.27a	14.17 ± 0.24b
Palmitoleic, C_16:1 ϖ-7_	0.05 ± 0.01a	0.06 ± 0.01a	1.94 ± 0.11b
Margaric, C_17:0_	0.00 ± 0.009a	0.05 ± 0.010a	0.00 ± 0.042a
Stearic, C_18:0_	2.03 ± 0.58a	2.81 ± 0.43a	4.16 ± 0.18b
Oleic, C_18:1 ϖ-9_	23.50 ± 1.20b	14.90 ± 1.58a	62.34 ± 1.07c
Linoleic, C_18:2 ϖ-6_	31.11 ± 2.10b	71.11 ± 3.08c	15.63 ± 1.57a
Gamma-linolenic, C_18:3 ϖ-6_	0.48 ± 0.07c	0.25 ± 0.04b	0.00 ± 0.02a
Alpha-linolenic, C_18:3 ϖ-3_	26.39 ± 0.89b	0.17 ± 0.09a	0.58 ± 0.04a
Gondoic, C_20:1 ϖ-9_	5.35 ± 1.11b	0.11 ± 0.08a	0.00 ± 0.02a
Eicosadienoic C_20:2 ϖ-6_	0.48 ± 0.06b	0.00 ± 0.09a	0.00 ± 0.01a

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
