# Peer review of "Comparison of the Oxidative Stability and Antioxidant Activity of Extra-Virgin Olive Oil and Oils Extracted from Seeds of Colliguaya integerrima and Cynara cardunculus under Normal Conditions and After Thermal Treatment"

_antioxidants, 2019, doi:10.3390/antiox8100470_

Round 1
Reviewer 1 Report
The publication Comparison of the Oxidative Stability and Antioxidant Activity of Extra-virgin Olive Oil and Oils Extracted From Seeds of Colliguaya integerrima and Cynara cardunculus Under Normal Conditions and After Thermal Treatment is interesting.
The obtained results are new and show the possibilities of using new sources of oil with potentially high oxidative stability.
The text is written correctly, however the authors should improve some parts.
The main comments.
In all the work there is no information on the statistical methods used by authors. However the authors write in the text that some results were significantly different and others were not.
Authors should,
Write about statistical method used in the research (in materials and methods part). The tables with results should include the mean, statistical deviation and significance of differences between the samples. Currently, based on the results, it is difficult to conclude on the existence or not of differences. The reviewer also doesn't understand the part (line 413-416):
“The GC-MS chromatograms obtained for CIO and CO before and after deep-frying were very different. However, when comparing CIO and CO oils between them after the heating process, they are indistinguishable. This result demonstrates that the heating process did not significantly affect the chromatographic profile of the secondary metabolites.”
Looking at figure 5, you can get the impression that there are more differences between the oils (CIO and CO) than between unheated and heated oil. So it is contrary to the text of authors.
The authors in this part write about secondary metabolites formed during heating, but the chromatograms in figure 5 (and the table S1 and S2) show a lot of substances, not only secondary metabolites. Maybe authors should write something about secondary metabolites and about their role in high temperature? The authors conclude that some of them may play an important role, but do not explain which of them and why. This part should be expanded / changed.
The frying process. The authors should explain and describe the heating process. There is no information about the time, mass of oil, the ratio between surface and volume the oil etc. This parameters can make a big impact on oil changes. The authors in the text write only the temperature. Authors sometimes use “the temperature of frying process” (probably this mean the temperature 180 °C), sometimes “after deep frying process”. The authors should unify these phrases, remembering that the frying process was not used in the experiment. The best way will be put the temperature of the heating process.
Oil extraction.
Do the authors have knowledge about the composition of the oils obtained by pressing, not by extraction? Would the nutritional value, composition of fatty acids and content of antioxidant compounds be the same?
Tables and figures The name of tables should be improved so that it can be independent content (without the text of papier). The same applies to the signature under the tables and explanations of the abbreviations in the table, and when the statistics appear, its explanations. In table 1 in TPC should be [mg gallic acid/kg oil] not Kg. In figure 5 it is better to show first the data from cold oil and after heat oil. Tables S1 and S2. The title should be change. The authors did not conduct the frying process. They heated the oil. Also gamma tocopherols and sitosterol have different letter (γ).
Author Response
Talca, September 26th, 2019
Manuscript ID: Antioxidants-590667
TITLE: Comparison of the Oxidative Stability and Antioxidant Activity of Extra-virgin Olive Oil and Oils Extracted From Seeds of Colliguaya integerrima and Cynara cardunculus Under Normal Conditions and After Thermal Treatment.
Dr. Molly Xia
Assistant Editor Antioxidants
Dear Dr. Molly Xia,
Please find below the point by point response, where we expect to satisfy the referees’ concerns and finally convince them of the value of this report. We also sincerely appreciate the opportunity for this new resubmission to Antioxidants.
All new additions and modifications of the article are highlighted in yellow.
Sincerely,
Dr. Oscar Valdés
Universidad Católica del Maule, Chile
Response to referee 1:
Regarding the comment: “Write about statistical method used in the research (in materials and methods part). The tables with results should include the mean, statistical deviation and significance of differences between the samples. Currently, based on the results, it is difficult to conclude on the existence or not of differences. The reviewer also doesn't understand the part (line 413-416):”
Response: Thanks for the advice. In this paper, we put only the mean values of all measurements, now in the new version of the manuscript, we added the standard deviation obtained by the ANOVA test according to the referee’s recommendation. In addition, we have included the statistical method used in the M&M section. Text highlighted in yellow.
Regarding the comment: “Looking at figure 5, you can get the impression that there are more differences between the oils (CIO and CO) than between unheated and heated oil. So it is contrary to the text of authors.”
Response: The referee was correct. Now in the new version this part was rewritten aiming at making it easier to understand. Text highlighted in yellow.
Regarding the comment: “The authors in this part write about secondary metabolites formed during heating, but the chromatograms in figure 5 (and the table S1 and S2) show a lot of substances, not only secondary metabolites. Maybe authors should write something about secondary metabolites and about their role in high temperature? The authors conclude that some of them may play an important role, but do not explain which of them and why. This part should be expanded / changed.”
Response: It was rewriting the text explaining the goal to identify secondary metabolites (changed to minor components throughout the text). It was now described that the study of the minor components in both oils (CIO and CO) were to obtain a characterization for future applications of CO as edible oil. Thus, the identification of the minor components in the studied oils using the GC-MS technique was performed. Please, see the changes in the new version of the manuscript. Text highlighted in yellow.
Regarding the comment: “ The frying process. The authors should explain and describe the heating process. There is no information about the time, mass of oil, the ratio between surface and volume the oil etc. This parameters can make a big impact on oil changes. The authors in the text write only the temperature. Authors sometimes use “the temperature of frying process” (probably this mean the temperature 180 °C), sometimes “after deep frying process”. The authors should unify these phrases, remembering that the frying process was not used in the experiment. The best way will be put the temperature of the heating process.”
Response: We agree. In order to improve the comprehension of the text, we have included the following text “It is important to note that the time and temperature were chosen taking into account the results reported by Mba et al., Andrikopoulos et al. and Naz et al., respectively”. It is important to note that all authors are leading researchers in the field of performance evaluation of natural and (semi)synthetic antioxidants under frying conditions. In addition, and taking into account his/her suggestion, we replaced the phrases such as, the temperature of the frying process, after the deep-frying process and others similar with heated at 180 °C, which is the temperature of the heating process. We believe that all these changes are intended to facilitate your understanding. Text highlighted in yellow.
Regarding the comment: “Do the authors have knowledge about the composition of the oils obtained by pressing, not by extraction? Would the nutritional value, composition of fatty acids and content of antioxidant compounds be the same?”
Response: At this moment we are unable to answer the question about the composition of the oils obtained by the pressing method. Despite this, it is known that the oil yield will be much lower in pressing than in solvent extraction. Moreover, the composition of some minor components, quality and the physicochemical properties (e.g. PV, MP, IV) of the oils could be influenced by the extraction process. Nevertheless, we believe that the fatty acid composition will be similar in oils obtained by both extraction systems. The results obtained by Ixtaina et al. (Ixtaina et al. Characterization of chia seed oils obtained by pressing and solvent extraction. Journal of Food Composition and Analysis. 24, 166–174, 2011) and Khattab et al. (Khattab et al. Quality evaluation of flaxseed oil obtained by different extraction techniques. LWT - Food Science and Technology. 53, 338-345, 2013) just to name two examples, support the aforementioned. We kindly invite the reviewer to accept the proof of concept that we are providing in the present form, which shows the correlation between the thermal stability, the phenolic content with the antioxidant activity.
Regarding the comment: “The name of tables should be improved so that it can be independent content (without the text of papier). The same applies to the signature under the tables and explanations of the abbreviations in the table, and when the statistics appear, its explanations. In table 1 in TPC should be [mg gallic acid/kg oil] not Kg. In figure 5 it is better to show first the data from cold oil and after heat oil. Tables S1 and S2. The title should be change. The authors did not conduct the frying process. They heated the oil. Also gamma tocopherols and sitosterol have different letter (γ).
Response: We thank the referees for the suggestions made to improve the wording, tables, and figures. Text highlighted in yellow.
Reviewer 2 Report
Dear Authors,
the paper must be strongly improved before possible publication in Antioxidants.
Line 22: EVOO instead of OVO
line 25: 420°C is too high for frying. Why this value? it is reported only in the abstract! I don't understand. The temperature of frying generally is 170-190 °C. The section 2.6 is very lacking. Is it a heating or a frying? there is a lot of confusion between the temperatures of heating, frying and gravimetric analysis (in the text!).
LINE 27: DPPH must be explained, but in my opinion at this point, DPPH is not necessary.
Line 31: GC/MS must be explained. Is it GC or HRGC?
Line 68: I suggest to the authors to put two figures, each for each plant (Colliguaya and Cynara) and with their seeds. In alternative to this proposal, I suggest to put these two figures (plants + seeds + obtained oils) at the beginning of the R&D section.
Line 94: FC is necessary?
Line 95: what is the color of the bottles? Where is EVOO from? I suggest you to put some information about EVOO sample (only one?)
line 114: delete s …oil
Lines 123-126: Absλ? Please, define it!
DL is D x L? or?
Wavelengths instead of wavenumbers
Line 108: are the seeds entire or grounded? if grounded, how is carried out?
Line 112-113: what is the standard deviation of the yields? Are they mean values of a number of extractions?
I suggest you to move the paragraph 2.4.3 after 2.4.6 or, better, after 2.4.7 because the values of the formula have been obtained with FA composition.
Line 158 delete s…acid
Line 191: I suggest you to delete DPPH. It is the reagent used for the assay. It must be in the 2.1. section
line 217: different temperatures?which??
Line 218: was the frying carried out only one time for each condition? It must be explained. Why the authors used 15 min that is a short time for a frying?
Line 228----Where is the section “Statistical analysis”? It is very important. In fact it is very serious that the values reported in table 1 and 2 are without SD (standard deviations) n=????
line 229: The R&D section referred to frying at ??? temperature?
Line 230: the abbreviations CIO, CO, EVOO were used previously, so it can be used, as such.
Line 234-236; table 1: the authors must improve the chemical name of the FA, for example oleic acid (C18:1 ϖ9). I don’t understand gamma- linolenic’s: is it a sum of isomers? The name as plural is uncorrected.
Caption of table 1: Chemical characteristics and fatty acid composition of…
Table 1: I suggest to put at first: Cox value, AV, …and then the FA composition. I strongly suggest to the author to control the limit values of the regulation of EVOO (2015/1830). For example, the AV for EVOO is 0.8, without other decimals. The same observation is valid for Table 2. The K value is with two decimals, not three.
Table 1: the values must have their standard deviation. This aspect is very lacking.
Table 2: I suggest you to move the data reported in table 2 into table 1, without the lines of the maximum permitted values (this aspect will be discussed in the text).
Line 230: the R&D discussion must start from chemical characteristics and yields of oil production. I suggest you to put two figures of the respective oils.
Figure 5: Why the authors did not report the EVOO, before and after frying? The discussion must be improved. It is very lacking. I strongly suggest you to use the references at this point to ameliorate the discussion:
- Eur. J. Lipid Sci. Technol., 116, 407-412, 2014; doi: 10.1002/ejlt.201300205 (volatiles of linoleic acid, which is the main FA of CIO and CO)
Figure 1-4: EVOO instead of OVO.
Figure 5: I suggest you to put before the oils before frying and after the oils after frying. Temperature of frying? I don't understand.
The conclusions must be improved.
Author Response
Talca, September 26th 2019
Manuscript ID: Antioxidants-590667
TITLE: Comparison of the Oxidative Stability and Antioxidant Activity of Extra-virgin Olive Oil and Oils Extracted From Seeds of Colliguaya integerrima and Cynara cardunculus Under Normal Conditions and After Thermal Treatment.
Dr. Molly Xia
Assistant Editor Antioxidants
Dear Dr. Molly Xia,
Please find below the point by point response, where we expect to satisfy the referees’ concerns and finally convince them of the value of this report. We also sincerely appreciate the opportunity for this new resubmission to Antioxidants.
All new additions and modifications of the article are highlighted in yellow.
Sincerely,
Dr. Oscar Valdés
Universidad Católica del Maule, Chile
Response to referee 2:
Regarding the comment: “Line 22: EVOO instead of OVO”
Response: Thanks for the advice. We have followed the referee´s recommendation, and now we changed EVOO for OVO. Text highlighted in yellow.
Regarding the comment: “Line 25: 420°C is too high for frying. Why this value? it is reported only in the abstract! I don't understand. The temperature of frying generally is 170-190 °C. The section 2.6 is very lacking. Is it a heating or a frying? there is a lot of confusion between the temperatures of heating, frying and gravimetric analysis (in the text!)”
Response: We agree. The values mentioned in Line 25: 420°C correspond to the final decomposition temperature of the studied oils obtained by gravimetric analysis. The value corresponding to the frying temperature is in the range of 170-190 °C, as you mentioned correctly. For this reason, and in order to explain the goal to identify secondary metabolites (changed to minor components throughout the text), we heated the studied oils at 180 °C for 30 min (the value that simulates the conditions of the frying process) and used the GC-MS technique to measure the minor components for each oil. So, now we have included in the abstract and in the M&M section (specifically section 2.6) the following text “…Moreover, according to the gravimetric analysis, the main decomposition step occurred in the temperature range of 200 - 420 ºC, showing a similar thermal behavior of EVOO oil……. Finally, we analyzed the minor components before and after heated at 180 ºC of CIO and CO by GC/MS using library search programs.”, and “ Two different heating processes were performed for the studied oils. The first heating process was performed to determine the total phenolic content and antioxidant activity of the oils for every selected temperature obtained in the thermal analysis. Specifically, 1 mL of 9 samples were added into a clean, acid-washed glass beaker (outer diameter about 18 mm, capacity 5 ml) and heated in a hot place with a thermometer to control the temperature. Each sample was subjected to the different heating temperatures (T 30ºC, Tint, T0, T 5%, and T10%) for 15 min. The second heating process was performed to measure the minor components present in the studied oils. In this case, we used the same methodology discussed above, but the samples were heated at 180 °C for 30 min. It is important to mention that this temperature value simulates the conditions of the frying process. After that, the samples were directly infused into the spectrometer before and after 180 ºC to determine the minor components using the GC-MS technique.”. In addition, we replaced phrases such as, the temperature of the frying process, after the deep-frying process and others similar with heated at 180 °C, which is the temperature of the heating process. We believe that all these changes are intended to facilitate your understanding. Text highlighted in yellow.
Regarding the comment: “Line 27: DPPH must be explained, but in my opinion at this point, DPPH is not necessary.”
Response: We have followed the referee's recommendation and the sentence has been corrected. Text highlighted in yellow.
Regarding the comment: “Line 31: GC/MS must be explained. Is it GC or HRGC?”
Response: The technique used for determining secondary metabolites (changed to minor components throughout the text), is GC-MS. In the article we now clarify these terms and rewrote this section with more details. Text highlighted in yellow
Regarding the comment: “Line 68: I suggest to the authors to put two figures, each for each plant (Colliguaya and Cynara) and with their seeds. In alternative to this proposal, I suggest to put these two figures (plants + seeds + obtained oils) at the beginning of the R&D section.”
Response: We fully agree with the referee and his/her suggestion has been included in the paper. Text and figure highlighted in yellow
Regarding the comment: “Line 94: FC is necessary?”
Response: No. In the text, we deleted the FC initial name word. Text highlighted in yellow.
Regarding the comment: “Line 95: what is the color of the bottles? Where is EVOO from? I suggest you to put some information about EVOO sample (only one?).
Response: The Extra-virgin olive oil (Olivares de Quepu) in dark green glass bottles was obtained from the local market and it is production come from the VII region of Chile. Text highlighted in yellow.
Regarding the comment: “line 114: delete s …oil”
Response: We agree. Text highlighted in yellow.
Regarding the comment: “Lines 123-126: Absλ? Please, define it!; DL is D x L? or?; Wavelengths instead of wavenumbers ”
Response: Thanks for the advice. We have followed the referee’s recommendation, and now the procedure for determining the specific extinction coefficient at different wavelengths was rewritten. Specifically, we have included the Absλ definition, replaced the wavenumbers and improved formula 1. Text highlighted in yellow.
Regarding the comment: “Line 108: are the seeds entire or grounded? if grounded, how is carried out?”
Response: The seeds used were grounded. In the article we now clarify this process and rewrote this section with more details. Text highlighted in yellow.
Regarding the comment: “Line 112-113: what is the standard deviation of the yields? Are they mean values of a number of extractions?”
Response: We agree. In this paper we put only the mean values of all measurements, now in the new version of manuscript we added the standard deviation according to the referee’s recommendation. Text highlighted in yellow.
Regarding the comment: “I suggest you to move the paragraph 2.4.3 after 2.4.6 or, better, after 2.4.7 because the values of the formula have been obtained with FA composition”
Response: Thanks for the advice. We have followed the referee´s recommendation, and now we moved paragraph 2.4.3 after 2.4.7. Text highlighted in yellow.
Regarding the comment: “Line 158 delete s…acid”
Response: We agree. Text highlighted in yellow.
Regarding the comment: “Line 191: I suggest you to delete DPPH. It is the reagent used for the assay. It must be in the 2.1. section”
Response: We agree. Text highlighted in yellow.
Regarding the comment: “line 217: different temperatures? which??
Response: The selected five temperatures chosen from the TG curves were mentioned in line 349 in the R&D section. Specifically, the T30, Tint, T0, T5%, and T10% were 30, 150, 250, 273 and 275 ºC, respectively. Despite this, in the new corrected version we mentioned this temperature. Text highlighted in yellow.
Regarding the comment: “Line 218: was the frying carried out only one time for each condition? It must be explained. Why the authors used 15 min that is a short time for a frying?
Response: We agree. As explained above, this section was rewritten aiming at making it easier to understand. The GC-MS chromatograms of the CIO and CO oils were taking before and after heating the oil to 180 ºC for 30 min. In order to improve the comprehension of the text, we have included the following “It is important to note that the time and temperature were chosen taking into account the results reported by Mba et al., Andrikopoulos et al. and Naz et al., respectively”. It is important to note that all authors are leading researchers in the field of performance evaluation of natural and (semi)synthetic antioxidants under frying conditions. Text highlighted in yellow.
Regarding the comment: “Line 228----Where is the section “Statistical analysis”? It is very important. In fact, it is very serious that the values reported in table 1 and 2 are without SD (standard deviations) n=????”
Response: Thanks for the advice. As commented above, in this paper we put only the mean values of all measurements, now in the new version of the manuscript we added the standard deviation obtained by the ANOVA test according to the referee’s recommendation. In addition, we have included the statistical method used in the M&M section. Text highlighted in yellow.
Regarding the comment: “line 229: The R&D section referred to frying at ??? temperature?”
Response: We do not agree with this comment because in this line nothing is said about frying temperature.
Regarding the comment: “Line 230: the abbreviations CIO, CO, EVOO were used previously, so it can be used, as such.
Response: We agree. Text highlighted in yellow.
Regarding the comment: “Line 234-236; table 1: the authors must improve the chemical name of the FA, for example oleic acid (C18:1 ϖ9). I don’t understand: is it a sum of isomers? The name as plural is uncorrected”.
Response: We fully agree with the referee’s comments and the corrections were made in the new version of the manuscript. Text highlighted in yellow.
Regarding the comment: “Caption of table 1: Chemical characteristics and fatty acid composition of…”
Response: We agree. Text highlighted in yellow.
Regarding the comment: “Table 1: I suggest to put at first: Cox value, AV, …and then the FA composition. I strongly suggest to the author to control the limit values of the regulation of EVOO (2015/1830). For example, the AV for EVOO is 0.8, without other decimals. The same observation is valid for Table 2. The K value is with two decimals, not three.”
Response: Thanks for the advice. We have followed the referee's recommendation and it has been corrected and included in the paper. Text highlighted in yellow.
Regarding the comment: “Table 1: the values must have their standard deviation. This aspect is very lacking.”
Response: This comment has already been resolved and discussed previously.
Regarding the comment: “Table 2: I suggest you to move the data reported in table 2 into table 1, without the lines of the maximum permitted values (this aspect will be discussed in the text).”
Response: We fully agree with the referee and his/her suggestion has been included in the paper. In the revised version we deleted Table 2 and the new text was highlighted in yellow.
Regarding the comment: “Line 230: the R&D discussion must start from chemical characteristics and yields of oil production. I suggest you to put two figures of the respective oils.”
Response: We agree with the referee’s comment and taking into account his/her suggestion, a new figure has been included in the paper as commented above. Text and figure highlighted in yellow
Regarding the comment: “Figure 5: Why the authors did not report the EVOO, before and after frying? The discussion must be improved. It is very lacking. I strongly suggest you to use the references at this point to ameliorate the discussion: - Eur. J. Lipid Sci. Technol., 116, 407-412, 2014; doi: 10.1002/ejlt.201300205 (volatiles of linoleic acid, which is the main FA of CIO and CO)
Response: The main goal of figure 5 (now Figure 6 in the text) was trying to identify the minor compounds present in CIO and CO oils. The phrase was re-written trying to improve the discussion of CG-MS chromatograms. When comparing CIO with CO oils, we observed that CO achieved less oxidation after heating, which is in accordance to its constituents and that CIO showed more oxidable linolenic acid in its composition (Table 1). It was pointed out in the new text, and it was explained in the text that the goal of GC-MS experiments were performed trying to understand the composition of the minor constituents of CIO and CO oils after heating processes. Text highlighted in yellow.
Regarding the comment “Figure 1-4: EVOO instead of OVO.”
Response: We agree. Text 3highlighted in yellow.
Regarding the comment: “Figure 5: I suggest you to put before the oils before frying and after the oils after frying. Temperature of frying? I don't understand.
Response: Thanks for the advice. As commented above, and following the referee´s recommendation in the new version of manuscript, we replaced phrases such as, the temperature of the frying process, after the deep-frying process and others similar with heated at 180 °C, which is the temperature of the heating process. We believe that all these changes are intended to facilitate your understanding. Text highlighted in yellow.
Regarding the comment: “The conclusions must be improved”.
Response: We agree with the referee’s comment and taking into account his/her suggestion, a new conclusion has been included in the paper with the new information provided. Text highlighted in yellow.
Final Comment: We thank the referees for the suggestions made to improve the wording and figures. We have made those corrections and improved the figures in size and layout.
Round 2
Reviewer 2 Report
Dear Authors,
the revised paper can be accepted for publication in Antioxidants, now.